# Evaluation of the Defined Bacterial Consortium Efficacy in the Biodegradation of NSAIDs

**DOI:** 10.3390/molecules28052185

**Published:** 2023-02-26

**Authors:** Ariel Marchlewicz, Urszula Guzik, Katarzyna Hupert-Kocurek, Danuta Wojcieszyńska

**Affiliations:** Institute of Biology, Biotechnology and Environmental Protection, Faculty of Natural Science, University of Silesia in Katowice, Jagiellońska 28, 40-032 Katowice, Poland

**Keywords:** NSAIDs, degradation, defined bacterial consortium, *Bacillus*, *Pseudomonas*

## Abstract

Due to the increasing pollution of wastewater with non-steroidal anti-inflammatory drugs, preparations need to be developed to decompose these drugs. This work aimed to develop a bacterial consortium with a defined composition and boundary conditions for the degradation of paracetamol and selected non-steroidal anti-inflammatory drugs (NSAIDs), including ibuprofen, naproxen, and diclofenac. The defined bacterial consortium consisted of *Bacillus thuringiensis* B1(2015b) and *Pseudomonas moorei* KB4 strains in a ratio of 1:2. During the tests, it was shown that the bacterial consortium worked in the pH range from 5.5 to 9 and temperatures of 15–35 °C, and its great advantage was its resistance to toxic compounds present in sewage, such as organic solvents, phenols, and metal ions. The degradation tests showed that, in the presence of the defined bacterial consortium in the sequencing batch reactor (SBR), drug degradation occurred at rates of 4.88, 10, 0.1, and 0.05 mg/day for ibuprofen, paracetamol, naproxen, and diclofenac, respectively. In addition, the presence of the tested strains was demonstrated during the experiment as well as after its completion. Therefore, the advantage of the described bacterial consortium is its resistance to the antagonistic effects of the activated sludge microbiome, which will enable it to be tested in real activated sludge conditions.

## 1. Introduction

Chemically, non-steroidal anti-inflammatory drugs (NSAIDs) belong to different groups but have a similar mechanisms of action. They are cyclooxygenase inhibitors involved in converting arachidonic acid to prostaglandins [1]. Their effectiveness in treating pain and inflammation translates into their popularity among patients. For example, it is estimated that the annual consumption of diclofenac in Germany is almost 90 tonnes, and in England, it is over 26 tonnes [2].

The beginning of research on levels of drugs, including NSAIDs in waters, soils, and sewage, dates back to the late 1990s, after publication of the Ternes report [3] describing the presence of 32 drugs and their metabolites in German rivers and streams.

Currently, water status monitoring is carried out for the presence of pharmaceuticals both in surface waters and outflows from wastewater treatment plants [4,5,6]. Data show that non-steroidal anti-inflammatory drugs are found in waters on all continents. Among other things, ibuprofen was identified at concentrations of up to 113 ng/L in the Pearl River (China), up to 4.35 g/L in the Sindian River (Taiwan), and up to 6.40 g/L in the rivers of Canada. In turn, high naproxen concentrations of 11.40–32.00 and 12.3 µg/L were observed in the rivers of Pakistan and Turkey, respectively. Characterized by high resistance to degradation, diclofenac has been identified in European rivers, such as in Poland (up to 0.429 µg/L), Spain (0.148 µg/L), and Ukraine (0.240 µg/L) [7,8,9]. This drug was even identified in streams in Antarctica on the Fildes Peninsula (84 ng/L) and Seymour/Marambio Island (77 ng/L) [10].

After ingestion, drugs are not entirely degraded but transformed via phase I and II biotransformations, and then excreted from the body in the parent form or the resulting metabolites. In stage I, oxidation, reduction, or hydrolysis most often occur, resulting in more hydrophilic compounds with active groups to which glucuronic acid, sulfates, or amino acids can attach in phase II. As a result of these reactions, hydroxyl and glucuronic derivatives are most often formed, leading to changes in the physicochemical properties of compounds. The consequence of these transformations is the observed impact on organisms living in the environment to which NSAID metabolites are discharged with sewage treatment plant effluents [11].

There are more and more reports on the impact of short- and long-term exposure to NSAIDs and their metabolites on non-target organisms [1]. The toxic effect of low concentrations of diclofenac have been shown on such species as carp (*Cyprinus carpio*), brown trout (*Salmo trutta fario*), rainbow trout (*Oncorhynchus mykiss*), and stickleback (*Gasterosteus aculeatus*) [12]. This drug can lead to histopathological changes in the liver and gills, cell oxidative stress, and even behavioral changes. Moreover, it has been shown that its degradation products, benzoquinone imines, can react with nucleophilic groups of proteins to form adducts [13]. In turn, Marchlewicz et al. [14] indicated that even ibuprofen, which is considered to be safe, could inhibit the reproductive ability of *Daphnia magna* at low concentrations and a dose of 0.4 µg/L ibuprofen inhibited the growth of duckweed. Chronic toxicity studies conducted with *Daphnia magna* showed that the amounts of ibuprofen at which the survival and development of this crustacean population were reduced were 8 and 2 µg/L (LOEC—lowest observed effect concentration), respectively [14,15].

Due to the widespread use of non-steroidal anti-inflammatory drugs, a systematic increase in their concentrations in the environment is predicted in the near future, especially over-the-counter drugs such as acetylsalicylic acid, paracetamol, ibuprofen, naproxen, or diclofenac [11]. Hence, more and more attention is being paid to methods of decomposing these compounds. Biological processes are among the cheapest and most environmentally friendly methods. Obtaining strains that can degrade NSAIDs remains challenging because these compounds have not been found in nature until recently, and their structure is resistant to microbial degradation. Thus, few microorganisms capable of completely degrading these bioactive compounds have been isolated so far. These include mainly bacteria of the genera *Pseudomonas, Bacillus, Rhodococcus, Stenotrophomonas, Citrobacter, Micrococcus, Rhizorhabdus, Labrys, Planococcus, Corynebacterium*, or *Enterococcus* [4,5,6,16,17,18,19,20,21,22,23]. The best-described strains include *Bacillus thuringiensis* B1(2015b), which is capable of degrading naproxen via salicylic acid and ibuprofen via 1,4-hydroquinone as critical metabolites, and *Pseudomonas moorei* KB4, which is able to degrade diclofenac and high concentrations of paracetamol [13,24,25,26]. Using the degradation potential of such strains requires the development of microbiological preparations that could support the work of sewage treatment plants in removing NSAIDs. Business entities responsible for the state of the waters are increasingly interested in such defined bacterial consortiums.

Due to the lack of effective microbiological preparations for the use of sewage treatment plants in the removal of NSAIDs, this research aimed to design a bacterial consortium with a strictly defined quantitative composition that would support municipal, pharmaceutical, and household sewage treatment plants in the removal of NSAIDs. Such preparations should be resistant to changes in external conditions and competition from the autochthonous microbiome of activated sludge. Hence, the second goal was to determine the scope of functioning of the bacterial consortium under changing environmental conditions and assess the survival of the introduced strains in the activated sludge, thereby ensuring the effectiveness of NSAID degradation in this system.

## 2. Results and Discussion

### 2.1. The Composition and Boundary Conditions for the Action of a Defined Bacterial Consortium

Due to the increasing consumption of NSAIDs, they more and more often end up in biological wastewater treatment plants where they are not always completely degraded. Hence, there is increasing demand for preparations enabling decomposition of these drugs in activated sludge. The barriers to success include the low survival rate of the introduced strains under activated sludge conditions and the low resistance of laboratory strains to changing conditions in the bioreactors of the sewage treatment plant. It is crucial to develop preparations based on strains that not only degrade NSAIDs but also show this ability under changing pH, temperature, and in the presence of other impurities [27].

#### 2.1.1. The Qualitative and Quantitative Composition of the Defined Bacterial Consortium

The candidates for constructing a bacterial consortium to support the removal of non-steroidal anti-inflammatory drugs from wastewater were the following strains: *Bacillus thuringiensis* B1(2015), capable of degrading ibuprofen and naproxen; *Pseudomonas moorei* KB4, capable of degrading paracetamol and diclofenac; and *Planococcus* sp. S5, capable of degrading naproxen and phenolic compounds [13,26,28,29]. In addition, we tested the possibility of introducing *Stenotrophomonas maltophilia* KB2, which degrades a wide range of aromatic compounds such as phenols, chlorophenols, cresols, and aromatic plant compounds, into the bacterial consortium [30,31,32,33]. The first stage of the research aimed to select the appropriate qualitative and quantitative ratios of the strains. The best quantitative ratios of the different bacterial strains were studied, determining the percentages of NSAID degradation and monitoring the cell count. Seven systems with varying proportions of strains B1(2015b):KB4:KB2:S5 were tested (Table 1). To identify the strains in the mixed culture, tested strains B1(2015b), KB4, and KB2 were labeled with fluorescent proteins (Figure 1). In turn, the S5 strain was marked with the rifampicin resistance gene. This made it possible to track quantitative changes in the strains of the tested systems.

As a result of the conducted tests, it was shown that the system that most effectively degraded the mixture of NSAIDs containing 10 mg/L paracetamol, 5 mg/L ibuprofen, 1 mg/L naproxen, and 1 mg/L diclofenac, was the system containing only strains B1(2015b) and KB4 in a ratio of 1:2 (Table 1). In addition, qualitative analysis of the composition of the tested systems showed that, although all of the examined strains were initially present in the S1 and S2 systems, the KB4 strain was not observed after 21 days of culture. After 35 days, the S5 strain was also not detected. In the S4 system, where the mixture of bacteria consisted of strains B1(2015b), KB4, and KB2 in a quantitative ratio of 1:1:1, the KB4 strain was also not observed on the 21st day due to the antagonism of the KB2 strain. At the same time, the KB2 strain has no adverse effect on the B1(2015b) strain that persisted in the culture. The KB2 strain was characterized by the fact that it did not survive in a system with a well-developed B1(2015b) culture. Hence, its presence was no longer observed on the 35th day. This allowed the population of the KB4 strain to recover, which was observed again on day 49. On the other hand, the B1(2015b) strain was below the detection limit at day 49, which was surprising and requires further research. In the S5 system in which the KB2 strain was absent, all of the introduced strains were still observed after 21 days, and only the presence of the S5 strain was not detected on the 35th day (Figure 2). These results indicated a negative impact of the KB2 strain on the KB4 strain and poor survival of the S5 strain in the mixed consortium.

The antagonistic effect of the KB2 strain on the KB4 strain likely resulted from the production of alkaline serine protease, which is responsible for the degradation of the Braun lipoprotein layer present in the outer lipopolysaccharide layer of Pseudomonas cells [34,35].

Due to the antagonistic effect of the KB2 strain against the KB4 strain, which was crucial in the degradation of paracetamol and diclofenac, and because the most effective degradation of the NSAID mixture was observed in the S7 system (Table 1), the bacterial consortium with strains B1(2015b) and KB4 in a ratio of 1:2 was selected for further research.

#### 2.1.2. Influence of Temperature and pH

The basic parameters influencing degradation processes in the environment are the temperature and pH of the environment. Preparations dedicated to supporting the operation of sewage treatment plants must operate at variable temperatures because the temperature in bioreactors changes seasonally, and the pH of sewage depends on the substances discharged from the sewage treatment plant [27]. Temperature plays a crucial role in degradation processes due to its influence on both bacterial physiology and the rate of enzymatic reactions. It is well known that most xenobiotic biodegradation processes occur at a temperature of 30–40 °C. In turn, the pH value affects the modification of the active site of degradation enzymes through its protonation or deprotonation, which changes the enzyme’s affinity for the substrate. In addition, the pH affects the charge on the surface of bacterial cells, which translates into the efficiency of sorption processes [26]. The bacterial consortium under development showed the highest efficiency in the temperature range of 20–35 °C (Figure 3a), which was related to the presence of mesophilic strains in the composition [26,29].

However, degradation of NSAIDs was also observed at 10 and 15 °C. Experiments on the influence of the environment’s pH on the degradation process of NSAIDs showed that the defined bacterial consortium had a wide range of activity from pH 5.5 to 9.0 (Figure 3b). Only ibuprofen degradation occurred in a narrower pH range from 6 to 8.5. These results were confirmed by the enzymatic activities described for the strains included in the bacterial consortium [24,25]. Górny et al. [24] and Marchlewicz et al. [15] indicated that the *Bacillus thuringiensis* B1(2015b) strain degraded naproxen and ibuprofen with the highest efficiency at 25–30 °C and pH 6–7.3. This was due to the activity of the main degradation enzymes involved in the decomposition of these drugs, such as catechol 1,2-dioxygenase, which has optimum activity at 25–30 °C and pH 7.3. In turn, phenolic monooxygenase, responsible for the first stage of hydroxylation of aromatic compounds, has optimum activity in the range of 25–30 °C and pH 8.0 [15]. by Żur et al. [26] reported that the second component of the defined bacterial consortium, the *Pseudomonas moorei* KB4 strain, degraded paracetamol most intensively at 30 °C and pH 7.0. However, intensive degradation was observed in the range from 15 to 40 °C. This strain has also shown high tolerance to a wide pH range [26]. In addition, the observed rapid decrease in the rate of NSAID degradation at pH 9 was probably correlated with a change in the charge on the cell surface. At high pH, cells show a negative charge [26]. Similarly, the negative form is taken by deprotonated drugs at high pH, which leads to negative changes in the interactions between drugs and the surface of bacterial cells, consequently inhibiting the degradation of NSAIDs.

#### 2.1.3. NSAID Degradation in the Presence of Co-Pollutants

Optimized NSAID degradation conditions were used to test drug degradation by the defined bacterial consortium (S7 system) over time. When testing the bacterial consortium under periodic conditions, it was shown that paracetamol was the fastest degraded NSAID in the mixture of drugs, the decomposition of which was 10 mg/L within the first 24 h. Additionally, ibuprofen was extensively degraded by the bacterial consortium, at 5 mg/L (Figure 4a). On the other hand, drugs containing two aromatic rings were degraded significantly more slowly, which confirmed previous data that the rate of decomposition depends on the degree of complexity of the chemical structure [13,24]. In addition, the poor degradation of diclofenac was probably related to the presence of chlorine atoms in its structure, which protect the aromatic ring from electrophilic attack during dioxygenase cleavage [36].

At the same time, during the degradation of the drug mixture, intensive growth of the KB4 strain was observed during the first 48 h, which was probably related to the degradation of paracetamol by this strain. On the other hand, the growth of strain B1(2015b) was more extended in time, which was related to the participation of this strain in the degradation of ibuprofen and naproxen. Qualitative and quantitative studies of the bacterial consortium’s composition during the degradation of the NSAID mixture showed that these strains did not act antagonistically and persisted in culture during the entire degradation process (Figure 4b).

Due to the heterogeneity of sewage in treatment plants, it is essential that co-contaminants do not completely inhibit the degradation activity of the bacterial consortium. Therefore, the degradation efficiency of the defined bacterial consortium (S7) was tested in the presence of common contaminants identified in wastewater. These experiments demonstrated that the tested bacterial consortium was highly effective in the presence of impurities such as ethanol, chromium, and lead. Moreover, compounds such as copper, phenol, acetone, or methanol did not inhibit the degradation of monocyclic NSAIDs (Figure 5).

The compound that strongly inhibited the bacterial consortium’s action, except for paracetamol degradation, was 2-nitrophenol (Figure 5f). It was previously shown that cytotoxic nitro compounds significantly (*p* < 0.05) inhibited the biodegradation of xenobiotics in soil and water as well as the activity of methanogenic consortia. This resulted in the inclusion of nitrophenols in the list of priority substances (HR-3 level) by the US Environmental Protection Agency, and it was recommended their concentrations be limited to levels below 10 ng/L in natural water reservoirs [37,38].

In the conducted study, a higher concentration of nitrophenol than the allowable level in the environment was used (1 mM or 139 mg/L), which led to over 90% inhibition of the degradation of ibuprofen, naproxen, and diclofenac. However, the high concentration of nitrophenol did not affect the rate of paracetamol degradation. This was likely because the bacterial breakdown of paracetamol occurred *via* a pathway involving aminophenol, a compound believed to be highly toxic to microorganisms [26]. In turn, nitro groups in biological systems can be reduced to amino groups with the participation of appropriate reductases. Although this process is most often observed in anaerobic systems, it has also been observed under aerobic conditions [39]. It can be assumed that the lack of inhibition of paracetamol degradation in the presence of nitrophenol was related to the high resistance of the *Pseudomonas moorei* KB4 strain to the toxic effects of aminophenol.

An interesting effect was observed in the presence of ethanol, which activated the decomposition of naproxen and diclofenac while inhibiting the degradation of ibuprofen. Ethanol is an activator of catechol 1,2-dioxygenase [40]. Górny et al. [24] showed that strain B1(2015b) cleaved the aromatic ring of naproxen via the ortho pathway, which explained the activating effect of ethanol. However, this strain cleaves the aromatic ring of ibuprofen using hydroquinol 1,2-dioxygenase, and the different degradation pathway of this drug was probably the reason for the inhibited degradation of this compound in the presence of alcohol [25].

In turn, the decomposition of paracetamol and ibuprofen was inhibited in the presence of copper ions. The toxic effect of copper ions on the decomposition of paracetamol was also noted by Żur et al. [26]. This effect was probably related to copper’s coordination of basic amino acids such as histidine in the active site of degradative enzymes. This precludes the attachment of ferrous ions to the active sites of hydroquinone 1,2-dioxygenase and hydroxyquinol 1,2-dioxygenase; thus, these are key ions in the aromatic ring cleavage of paracetamol and ibuprofen, respectively [41]. In addition, acetone was also a strong inhibitor of paracetamol degradation (Figure 5). This was probably also related to the interaction of this inhibitor with the active site of dioxygenase, which is involved in the cleavage of the aromatic ring of paracetamol. Bertini et al. [42] showed that aliphatic ketones had a high affinity for iron in the second oxidation state and that hydrophobic interactions played a crucial role in binding the inhibitor to the active site.

### 2.2. Degradation Studies in an SBR Bioreactor System

Because batch cultures in a mineral medium and without competing microorganisms do not fully reflect the real conditions in sewage treatment plants, mainly due to the synergistic metabolic activity exhibited by microorganisms in communities [27], further tests were conducted with the defined bacterial consortium in a sequencing batch reactor (SBR) system. An activated sludge culture was introduced into the reactor and fed with synthetic sewage containing a mixture of NSAIDs. In this arrangement, the bacterial consortium was observed to completely degrade 10 mg/L paracetamol within 24 h in each bioreactor cycle. Ibuprofen was degraded in each bioreactor cycle. However, each subsequent cycle slightly prolonged the degradation time of the next dose (5 mg/L). The first dose was eliminated after 24 h, but the next doses were removed within 3–5 days from the day of supplementation. Due to the long degradation time, naproxen and diclofenac were not completely eliminated during the seven day cycle of the SBR bioreactor. With the start of subsequent cycles, naproxen and diclofenac were added to the system so that the final concentration in the system was 1 mg/L. For both drugs, an improvement in drug degradation efficiency was observed with subsequent cycles (Figure 6a). Naproxen was degraded 33.80%, 46.30%, 53.57%, and 52.34% in subsequent cycles, while diclofenac was degraded 9.00%, 28.00%, 14.71%, and 35.58%. The maximum degradation rate was 0.1 mg/day for naproxen and 0.05 mg/day for naproxen and diclofenac, respectively.

During the degradation of drugs, the condition of the activated sludge was monitored by measuring the content of ammonium, nitrites, and nitrates. The introduced defined bacterial consortium and the drug degradation processes did not negatively affect the nitrification process. The rate of phenol degradation by the activated sludge microbiome was adopted as an indicator of the quality of activated sludge functioning. No disturbance of this process was observed (Figure 6b).

Contrary to the study by Żur et al. [27] and Michalska et al. [43], no intensification of phenol and nitrogen metabolism was observed after the introduction of artificial wastewater containing pharmaceuticals. Michalska et al. [43] indicated that the observed disturbances were related to the metabolic specialization of activated sludge microorganisms in the presence of introduced sewage with the simultaneous loss of functionally essential microorganisms. The lack of changes in nitrification and phenol decomposition processes after augmentation and supplementation with drugs in subsequent reactor cycles indicated that the applied treatments likely did not disturb the balance of the activated sludge microbial community.

In the first week, the number of bacteria in the continuous system (SBR bioreactor) ranged from 10^6^ to 10^10^ colony-forming units/mL (CFU/mL). In the next 3 weeks, the number of introduced bacteria remained constant for the B1 strain at 10^7^ CFU/mL and the KB4 strain at 10^9^ CFU/mL. The obtained values of drug degradation and strain survival in the bioreactor system, designed to simulate the operation of a biological sewage treatment plant, indicated the possibility of using the defined bacterial consortium in sewage treatment plant systems.

To date, no such functioning system has been described. Similar research by Żur et al. [27] was not successful. Despite using pharmaceuticals at lower concentrations, these authors did not observe a significant improvement in the intensity of drug degradation [27]. The use of the *Stenotrophomonas maltophilia* KB2 strain in the consortium, with a strong antagonistic effect against the KB4 strain, likely prevented improvement of the drug degradation efficiency. An optimized bacterial consortium has now been proposed as a drug bioremediation tool.

## 3. Materials and Methods

### 3.1. The Composition and Boundary Conditions for the Action of a Bacterial Consortium

*Bacillus thuringiensis* B1(2015b) (GenBank Accession Number KP895873.1), *Pseudomonas moorei* KB4 (GenBank Accession Number GCA_004212425.1), *Planococcus* sp. S5 (GenBank Accession Number AY028621.1), and *Stenotrophomonas maltophilia* KB2 (VTT E-113197) were routinely cultivated in nutrient broth (BBL^®^ Nutrient broth, Becton Dickinson, ND, USA) at 30 °C and 130× *g* for 24 h [26,29,31]. Subsequently, cells were harvested by centrifugation (5000× *g* at 4 °C for 15 min), washed with a fresh sterile mineral salts medium (MSM) [30], and used in further experiments.

### 3.2. Assessment of the Qualitative and Quantitative Composition of Microorganisms in Cultures

To determine the optimal composition of the bacterial consortium for degrading NSAIDs, culture systems were established with different ratios of tested and labeled strains (Table 1) in MSM [30] supplemented with 0.5 g/L glucose and drugs at a concentrations of 10 mg/L paracetamol, 5 mg/L ibuprofen, and 1 mg/L naproxen and diclofenac. Cultures were carried out at 30 °C and pH 7.3 for 49 days. During this time, the species composition of the culture and the concentrations of drugs were determined every 7 days.

#### 3.2.1. Fluorescent Labeling of *Stenotrophomonas maltophilia* KB2 and *Pseudomonas moorei* KB4 Strains

Electrocompetent cells of the KB2 and KB4 strains were prepared according to a standard protocol [44] with minor modifications. Briefly, 5 mL of overnight culture was inoculated in 100 mL of LB and grown with shaking at 30 °C until it reached 0.5 OD_600_. Cells were cooled on ice and centrifuged at 5000 rpm for 10 min at 4 °C. The pellet was resuspended in 100 mL and washed twice with 50 mL and then 25 mL of ice-cold sterile water. After washing, cells were resuspended in 2 mL of 10% glycerol and aliquoted into multiples of 50 µL. Electrocompetent cells of the KB2 and KB4 strains were transformed with the pMP4566 plasmid vector for the constitutive expression of the egfp gene and the pMP4662 vector for the constitutive expression of the rfp (DsRed) gene, respectively [45]. A 50 µL aliquot of electrocompetent cells was mixed with 5 μL of plasmid DNA (69.22 ng/μL) and incubated for 10 min on ice. After incubation, bacteria were transferred into pre-chilled electroporation cuvettes and electroporated using a Gene-Pulser (Bio–Rad Laboratories, Hercules, California, United States) set at 2.5 kV, 200 Ω, with a time constant of 4.5–5.4 ms. Immediately after electroporation, 0.95 mL of Luria-Bertani broth (LB) was added to the transformed cells and the bacteria were incubated at 30 °C with shaking for 45 min. After incubation, aliquots of 50, 100 and 300 μL of bacteria were plated on LB solid medium supplemented with 50 µg/mL tetracycline and incubated overnight at 30 °C. The resulting colonies were analyzed for bacteria fluorescence. Bacteria were examined using a Nikon ECLIPSE-Ni-U stereomicroscope with epifluorescence detection, using 480/40 nm excitation and a 510 nm long-pass emission filter for EGFP and 510/20 nm excitation with a 560/40 nm emission filter for DsRed.

#### 3.2.2. Fluorescent Labeling of *Bacillus thuringiensis* B1(2015b) Strain

Electrocompetent cells of the B1 strain were prepared according to Mahillon and Lereclus [46]. Next, electrocompetent cells were transformed with plasmid pAD43-25 carrying the functional gfp gene, enabling high-level constitutive expression of green fluorescence protein GFP (Bacillus Genetic Stock Center (BGSC), Department of Microbiology, Ohio State University, Columbus, OH, USA). A 100 µL aliquot of electrocompetent cells was mixed with 2 μL plasmid DNA (74.32 ng/μL). The same procedure for *Pseudomonas moorei* KB4 was then followed. However, after incubation of the transformed cells, aliquots of 100, 200 and 300 μL of bacteria were plated on LB solid medium supplemented with 25 µg/mL chloramphenicol and incubated overnight at 30 °C. The resulting colonies were analyzed for the presence of green fluorescent protein. For this purpose, bacteria were observed using a Nikon ECLIPSE-Ni-U stereomicroscope with epifluorescence detection, with 480/40 nm excitation and a 510 nm long-pass emission filter.

#### 3.2.3. Selection of a Rifampicin-Resistant Mutant of *Planococcus* sp. Strain S5

To monitor the cell count of the *Planococcus* sp. strain S5, spontaneous mutants of the strain were obtained by plating on a solidified LB medium supplemented with 2 μg/mL of rifampicin. The growing colonies were then reinoculated on LB plates supplemented with successively higher levels of rifampicin (up to 150 μg/mL of medium). The stability of the mutation was confirmed by culturing the mutant five times on LB medium without antibiotic selection [47].

### 3.3. NSAID Degradation Experiments

#### 3.3.1. Periodic Systems

Periodic cultures were carried out in MSM [30]. The cultures were cultivated at 30 °C. Glucose at a concentration of 0.5 g/L was used as the carbon source. A mixture of drugs was introduced into the culture: 5 mg/L ibuprofen, 1 mg/L naproxen, 1 mg/L diclofenac, and 10 mg/L paracetamol. To determine the optimum temperature for the degradation process, tests were carried out in the range of 4–40 °C. To determine the optimum pH for the biodegradation of drugs, experiments were carried out in the pH range of 5–9. To determine the impact of co-pollutants detected in the wastewater on NSAID biodegradation, tests were carried out in the presence of 2-nitrophenol, sodium benzoate, and phenol at a concentration of 1 mM; ethanol, methanol, and acetone at 1%; and copper (II), chromium (VI), and lead (II) ions at a concentration of 0.1 mM. Samples were taken from the cultures at specified intervals, and the concentrations of NSAIDs were determined. All experiments were performed in triplicate.

#### 3.3.2. Bioreactor System

The experiment was carried out in a Sartorius BIOSTAT A laboratory bioreactor with automatic control of oxygen content, pH, and temperature. The bioreactor with a working volume of 5 L was filled with the following:

-1000 mL of activated sludge from the aerated activated sludge chamber (Klimzowiec wastewater treatment plant, Chorzów, Poland);-2000 mL of synthetic sewage containing (per 1000 mL): 0.04 g NH_4_Cl, 0.024 g K_2_HPO_4_, 0.008 g KH_2_PO_4_, 0.1 g CaCO_3_, 0.2 g MgSO_4_ × 7 H_2_O, 0.04 g NaCl, and 0.005 g iron FeSO_4_ × 7 H_2_O [48], which was previously sterilized for 15 min at 121 °C;-0.6 mL of 30% glucose solution and ammonium acetate (0.317 g/L) sterilized by microfiltration;-NSAIDs: 5 mg/L ibuprofen sodium, 10 mg/L paracetamol, 1 mg/L naproxen sodium, and 1 mg/L diclofenac sodium;-co-contaminants: phenol (1 mM), methanol (1%), and Cu^2+^ (0.1 mM as Cu(NO_3_)_2_);-supplement: cow’s milk at a concentration of 5% (200 mL of milk per system).

Saturation was assumed to be maintained at 60%. One bioreactor cycle lasted 7 days. After this time, the activated sludge was sedimented and 2 L of the overlying liquid was removed and supplemented with fresh synthetic sewage containing all the components together with co-contaminants, thus achieving the assumed initial concentrations. Before adding the mixture of drugs, their levels in the system were analyzed and supplemented so that the final concentration in the bioreactor reached the initial values.

NaOH/HCl solutions (2%) with an accuracy of 0.1 units were used to control the pH. The process temperature was set at 18 °C. Samples were obtained from the culture every 24 h, and the concentrations of NSAIDs, phenol, ammonium ion, nitrite (III), and nitrate (V) were determined. All experiments were performed in triplicate.

### 3.4. Biochemical Analysis

The ammonium ion content was determined using the Nessler method [49]. For this purpose, the collected samples were centrifuged at 14,000 rpm for 15 min. A 0.5 mL aliquot of the sample was diluted to 5 mL with distilled water, and 0.5 mL of Nessler’s reagent was added. Absorbance was measured at λ = 410 nm after 10 min of incubation. Determination of the nitrites (III) content was carried out using the colorimetric method with naphthylamine [50]. The test sample was filtered using syringe filters with a pore diameter of 0.45 µm. A 0.5 mL volume of sulfanilic acid was added to 50 mL of sample and incubated for 5 min at room temperature. Then, 0.5 mL of naphthylamine and 0.5 mL of sodium acetate were added. The absorbance was measured at λ = 543 nm. Determination of the nitrates (V) content was carried out using the brucine method, and absorbance was measured at λ = 410 [51]. The phenol concentration was determined using a colorimetric method with p-nitroaniline at λ = 550 [52].

### 3.5. Determination of NSAIDs Concentration

The decomposition of NSAIDs was monitored by HPLC (Nexera LC-40, Shimadzu, Japan) in reverse phase equipped with a ReproSil-Pur Basic C-18 column (150 mm × 4.6 mm, 5 µm, Dr Maisch HPLC GmbH, Ammerbuch, Germany) and a fluorescent detector (for ibuprofen, diclofenac, naproxen) or PDA detector (for paracetamol). Separation was carried out using an isocratic flow of 1 mL/min. The obtained samples were centrifuged at 14,000 rpm for 20 min prior to HPLC analysis. The mobile phase in the determination of ibuprofen, naproxen, and diclofenac concentrations consisted of acetonitrile and 1% acetic acid (70:30 *v*/*v*). The detection wavelength was set: absorption—224 nm, emission—295 nm for ibuprofen; absorption—254 nm, emission—352 nm for diclofenac; absorption—286 nm, emission—360 nm for naproxen. The mobile phase in the determination of paracetamol consisted of acetonitrile and 1% acetic acid (10:90 *v*/*v*). The detection wavelength was set at 240 nm.

### 3.6. Statistical Analysis

Statistical analysis was performed using STATISTICA 13 PL software (TIBCO Software Inc., Palo Alto, CA, USA) based on one-way ANOVA tests with the significance level set at *p* < 0.05.

## 4. Conclusions

Bioaugmentation of activated sludge with bacterial strains with increased degradative capabilities is a promising solution for the removal of poorly biodegradable pharmaceuticals such as paracetamol, ibuprofen, diclofenac, and naproxen. In this research, a defined bacterial consortium dedicated to decomposing selected NSAIDs was developed, and boundary conditions for its action were developed. The introduction of the developed bacterial consortium composed of *Bacillus thuringiensis* B1(2015b) and *Pseudomonas moorei* KB4 strains in a 1:2 ratio did not change the nitrifying activity of the activated sludge’s microorganisms, but it increased their ability to degrade NSAIDs. The analyses showed that the mixed culture of microorganisms degraded ibuprofen at a rate of 4.88 mg/day, paracetamol at 10 mg/day, naproxen at 0.1 mg/day (0.7 mg/week), and diclofenac at a rate of 0.05 mg/day (0.34 mg per week). The species composition analysis showed the survivability of the introduced strains in the bioreactor system with a mixed population of activated sludge. The obtained results support testing the designed defined bacterial consortium in the real environment of a sewage treatment plant.

## Figures and Tables

**Figure 1 molecules-28-02185-f001:**
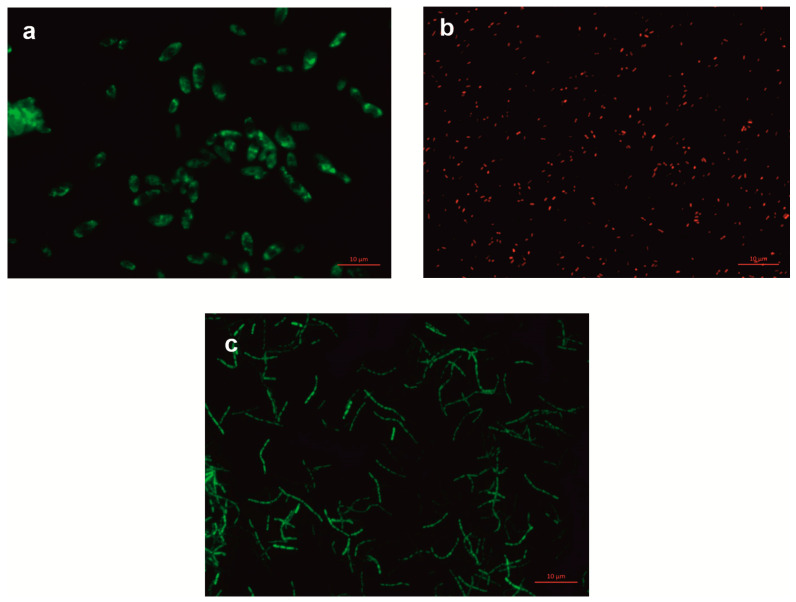
Fluorescence microscopy images of bacterial cells at 100× magnification. (**a**) *Bacillus thuringiensis* B1(2015b), (**b**) *Pseudomonas moorei* KB4, (**c**) *Stenotrophomonas maltophilia* KB2.

**Figure 2 molecules-28-02185-f002:**
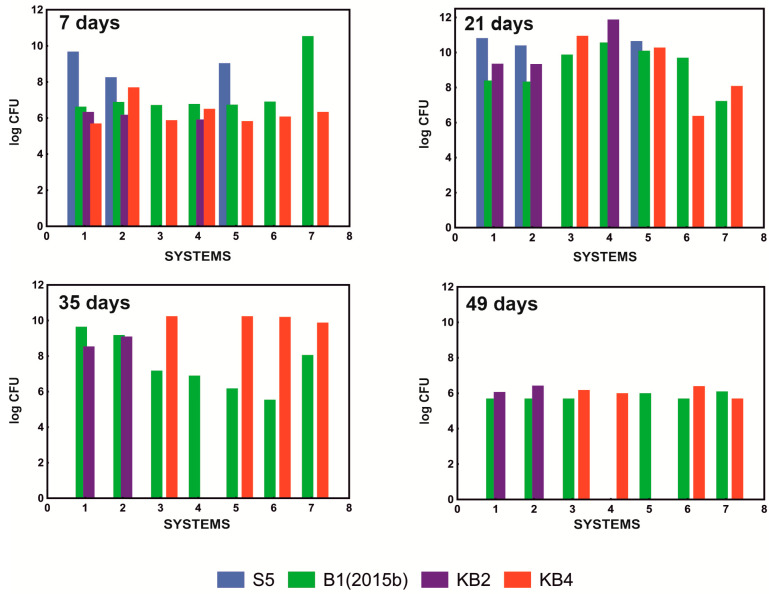
Survival of the tested strains in examined systems S1–S7 after 7, 21, 35, and 49 days of cultivation at pH 7.3 and 30 °C.

**Figure 3 molecules-28-02185-f003:**
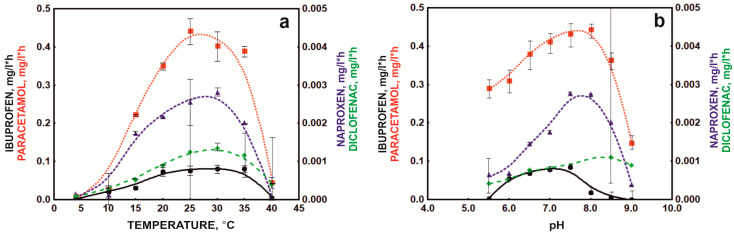
Influence of temperature (**a**) and pH (**b**) on NSAID degradation by the bacterial consortium.

**Figure 4 molecules-28-02185-f004:**
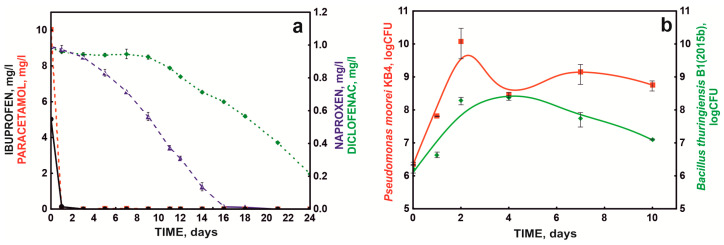
NSAID degradation (**a**) by the defined bacterial consortium (S7) and bacterial growth (**b**) in periodic culture under optimal conditions.

**Figure 5 molecules-28-02185-f005:**
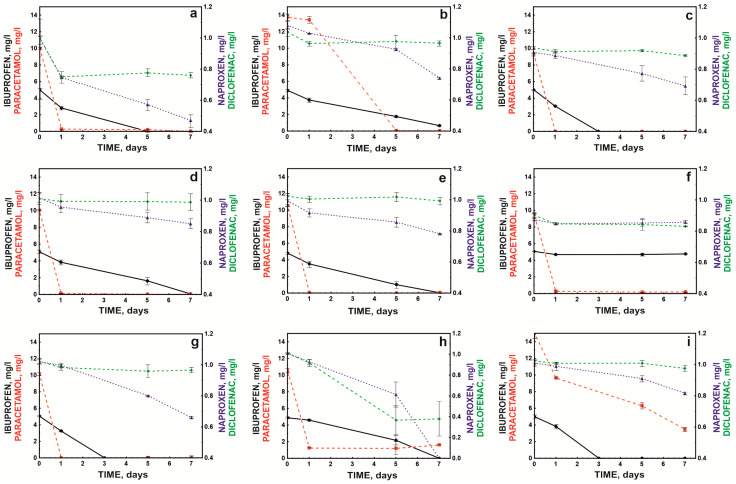
Influence of co-pollutants: Cr^+6^ (**a**), Cu^+2^ (**b**), Pb^+2^ (**c**), phenol (**d**), benzoate (**e**), 2-nitrophenol (**f**), methanol (**g**), ethanol (**h**), and acetone (**i**) on NSAID degradation by the defined bacterial consortium.

**Figure 6 molecules-28-02185-f006:**
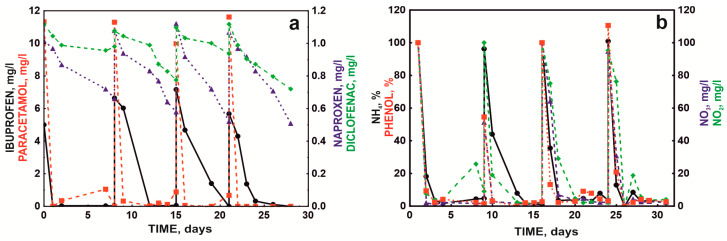
NSAID degradation by the defined bacterial consortium in the SBR bioreactor (**a**) and concentrations of phenol NH_4_^+^, NO_3_^−^, and NO_2_^−^ ions throughout the process (**b**) at pH 7.3 and 18 °C.

**Table 1 molecules-28-02185-t001:** Degradation of NSAIDs (in percentages) by bacterial consortia in defined proportions within 40 days, pH 7.3, temperature 30 °C.

System	Proportion of B1(2015b):KB4:KB2:S5	Paracetamol 10 mg/L	Ibuprofen 5 mg/L	Naproxen 1 mg/L	Diclofenac 1 mg/L
S1	1:1:1:1	100 ± 0.00	100 ± 0.00	16.30 ± 5.10	16.25 ± 3.39
S2	2:2:1:1	100 ± 0.00	100 ± 0.00	43.15 ± 4.29	2.05 ± 9.40
S3	1:1:0:0	100 ± 0.00	100 ± 0.00	50.90 ± 2.67	32.68 ± 0.25
S4	1:1:1:0	100 ± 0.00	100 ± 0.00	53.32 ± 0.89	10.75 ± 1.33
S5	1:1:0:1	100 ± 0.00	100 ± 0.00	49.26 ± 4.53	17.55 ± 3.32
S6	2:1:0:0	100 ± 0.00	100 ± 0.00	41.64 ± 3.04	40.76 ± 3.04
S7	1:2:0:0	100 ± 0.00	100 ± 0.00	74.02 ± 1.83	28.99 ± 4.25

## Data Availability

The data presented in this study are available upon request from the corresponding author. The data are not publicly available due to privacy restrictions.

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
