# Peer review of "Evaluation of the Defined Bacterial Consortium Efficacy in the Biodegradation of NSAIDs"

_molecules, 2023, doi:10.3390/molecules28052185_

Round 1

Reviewer 1 Report

Research is aimed at solving the acute problem of environmental pollution with pharmaceuticals, one of the most dangerous groups of emerging pollutants, which since the beginning of the 2000s have been considered a new class of micropollutants.

The manuscript has a pronounced applied character and describes the vaccine for the biodegradation of NSAIDs, which is resistant to the antagonistic effects of the activated sludge microbiome and substances polluting wastewater from wastewater treatment plants.

Specific comments:

Lines 68-69. It is recommended to expand references to the few (single) literature sources regarding the description of microorganisms capable of complete decomposition of NSAIDs. For example, 10.1007/s00253-012-4170-5, 10.1007/s00284-021-02543-4, 10.1016/j.ibiod.2022.105490, 10.1134/S0003683806040090 (paracetamol); 10.1038/s41598-019-45732-9, 10.1016/j.ecoenv.2018.01.040 (diclofenac), 10.1007/s12210-022-01085-6, 10.1007/s10123-022-00248-7, 10.1371/journal.pone.0260032, 10.1128/aem.00388-22, 10.1007/s13762-019-02400-9 (ibuprofen).

Lines 30–33: Please add a citation here.

The data presented (figures and tables) should include statistical analysis (at least SD). Lines 205-206: "It was shown that cytotoxic nitro compounds significantly (add a p value here) inhibited…". Line 211: same here.

Author Response

Dear Reviewer 1,

Thank you very much for your work and valuable comments. We have corrected the manuscript according to your suggestions. All introduced changes are marked in the file through the change tracking function. Moreover, all changes and explanations are listed below.

We will be very grateful if you accept our corrections.

Sincerely,

Urszula Guzik and other Co-authors

Reviewer comment

Lines 68-69. It is recommended to expand references to the few (single) literature sources regarding the description of microorganisms capable of complete decomposition of NSAIDs. For example, 10.1007/s00253-012-4170-5, 10.1007/s00284-021-02543-4, 10.1016/j.ibiod.2022.105490, 10.1134/S0003683806040090 (paracetamol); 10.1038/s41598-019-45732-9, 10.1016/j.ecoenv.2018.01.040 (diclofenac), 10.1007/s12210-022-01085-6, 10.1007/s10123-022-00248-7, 10.1371/journal.pone.0260032, 10.1128/aem.00388-22, 10.1007/s13762-019-02400-9 (ibuprofen).

Answer

Corrected as required.

Reviewer comment

Lines 30–33: Please add a citation here.

Answer

Corrected as required.

Reviewer comment

The data presented (figures and tables) should include statistical analysis (at least SD). Lines 205-206: "It was shown that cytotoxic nitro compounds significantly (add a p value here) inhibited…". Line 211: same here.

Answer

Standard deviations have been introduced in the table and figures. In lines 205-206, information about "(p<0.05)" was entered, while in line 211, the word "significantly" was removed because the information referred to literature data and no statistical analysis was performed here.

Reviewer 2 Report

Manuscript Molecules 2211802

This manuscript describes the selection of bacterial strains, that previously had shown capacity to degrade certain NSAIDs separately, to prepare a bacterial consortium capable of degrading 4 NSAIDs under activated sludge conditions. It is indeed a very comprehensive study worth to be published in Molecules, but making changes and improvements in some points.

The first change should be made in the title of the manuscript, because what the authors are using is not a microbial vaccine, but a bacterial consortium, and this change should be done in the title. A vaccine is defined as “a biological preparation that provides active acquired immunity to a particular infectious or malignant disease”. But, although the consortium used is a biological preparation, its role is not to fight against a disease or infection. Make the same change in keywords and in all the manuscript.

Besides, the manuscript seems to be disordered in some parts, as in Materials and Methods section. In general, there is a lack of information in the legends of figures and tables.

Here are some suggestions for improving the manuscript:

Abstract: Define SBR (Sequencing Batch Reactor) here.

Line 81: The survival of introduced strains is not evaluated in the activated sludge conditions, but only in some systems prepared in aqueous solution under changing environmental conditions. Correct.

2.1.1. The qualitative and quantitative composition of the consortium should be separated:

Line 102. …qualitative and quantitative ratios of the strains. (Add something as follows) “The best quantitative ratios of the different bacterial strains were studied determining the percentages of NSAIDs degradation and monitoring the cells counts”. Seven systems with varying proportions…(Table 1). To identify the strains…resistance genes.

Line 109: Breeding system?

Table 1. Give the conditions of pH and temperature used. What about the degradation over time?

Figure 2. Put in the figures 7 days, 21 days, 35 days, and 49 days instead of a, b, c, and d for an easier visualization of the figure. Give the conditions of pH and temperature used. In the system S4, how can it be explained that after 35 days KB2 is absent and B1 is present and after 49 days it is just the contrary?

Line 150. Where is the information that these are mesophilic strains?

Lines 175-183. This paragraph should be positioned after Table 1 and before the influence of temperature and pH, since it does not belong to the degradation in the presence of co-pollutants (2.1.3.).

Figure 4. Where do data in figure 4 come from? Which system? Conditions of pH and temperature? I think that data in figure 4a are given as percentages, and not as mg/l. Taking this into account, data in this figure after 24 days do not match those in Table 1 after 40 days.  In none of the systems studied in Table 1 the percentage of naproxen degraded reached 100%, and for dichlofenac, the maximum degradation reached after 40 days was 40.76%, but in figure 4a it reached 80% in 24 days.

Figure 5. According to Materials and methods section, the initial amounts of drugs used were: naproxen and dichlofenac 1 mg/l, ibuprofen 5 mg/l and paracetamol 10 mg/l, but at time 0 days these amounts are different in each of the different figures (a-i). Explain that.

I think that is necessary to insert a figure with data in the absence of the co-pollutants.

Line 205. …except for paracetamol.

Line 224. …naproxen´s and dichlofenac´s decomposition…

Lines 230-231….”the decomposition of paracetamol and ibuprofen were inhibited only in the presence of copper”. That is not true for ibuprofen, which is inhibited also by CR, Cu, phenol, benzoate and ethanol.

Line 242. 2.2. Degradation studies in a SBR bioreactor system

Figure 6. NSAIDs degradation by bacterial consortia in SBR bioreactor (a) and values of…..throughout the process (b).

Materials an Methods should be written in order, beginning with the degradation experiments and giving details about in which samples are applied the different techniques.

Line 3.3.1. …and stenotrophomonas maltophilia KB2?

Degradation experiments

In which experiments are the fluorescent bacteria used?

Only when the conditions have been selected?

Have you a parallel culture with and withot fluorescence?

4-chlorophenol, DMSO and ZnII have not been used as co-pollutent.

Author Response

Dear Reviewer 2,

Thank you very much for your work and valuable comments. We have corrected the manuscript according to your suggestions. All introduced changes are marked in the file through the change tracking function. Moreover, all changes and explanations are listed below.

We will be very grateful if you accept our corrections.

Sincerely,

Urszula Guzik and other Co-authors

Reviewer comment

The first change should be made in the title of the manuscript, because what the authors are using is not a microbial vaccine, but a bacterial consortium, and this change should be done in the title. A vaccine is defined as “a biological preparation that provides active acquired immunity to a particular infectious or malignant disease”. But, although the consortium used is a biological preparation, its role is not to fight against a disease or infection. Make the same change in keywords and in all the manuscript.

Answer

Thank you for this comment. It was taken into account in the manuscript.

Reviewer comment

Besides, the manuscript seems to be disordered in some parts, as in Materials and Methods section. In general, there is a lack of information in the legends of figures and tables.

Answer

In Figure 2, the signature of individual elements has been changed. Experimental days are entered directly into the panel figures to facilitate analysis of the results. No legend was introduced in the remaining figures because the individual series were distinguished by colours (results for naproxen are in blue, diclofenac - green, paracetamol - red, ibuprofen - black). The same range of colours has been used in all diagrams except for Fig. 4b. This was also indicated in the descriptions of the axes, where the names of drugs correspond to the colours of the data on the chart. Similarly, In Figure 4b, the data for Pseudomonas moorei KB4 (red) and Bacillus thuringiensis B1(2015b) (green) strains are coloured. In addition, the descriptions under the drawings have been extended with additional information. Materials and Methods section hass been also corrected.

Reviewer comment

Abstract: Define SBR (Sequencing Batch Reactor) here.

Answer

Corrected as required

Reviewer comment

Line 81: The survival of introduced strains is not evaluated in the activated sludge conditions, but only in some systems prepared in aqueous solution under changing environmental conditions. Correct.

Answer

Strain survival analysis was also performed in activated sludge, and the study results are presented and discussed in the manuscript in lines 318-324.

Reviewer comment

2.1.1. The qualitative and quantitative composition of the consortium should be separated:

Answer

The separation of the qualitative and quantitative analysis of the tested bacterial consortia is not possible because the changes in the tested systems each time were both qualitative and quantitative.

Reviewer comment

Line 102. …qualitative and quantitative ratios of the strains. (Add something as follows) “The best quantitative ratios of the different bacterial strains were studied determining the percentages of NSAIDs degradation and monitoring the cells counts”. Seven systems with varying proportions…(Table 1). To identify the strains…resistance genes.

Answer

Corrected as required.

Reviewer comment

Line 109: Breeding system?

Answer

The unfortunate phrase "breeding systems" has been changed to "cultures".

Reviewer comment

Table 1. Give the conditions of pH and temperature used. What about the degradation over time?

Answer

The tests were carried out at pH 7.3 and 30 degrees Celsius. This information is included in the description of Table 1. Degradation over time by a defined bacterial consortium under optimal conditions is shown in Figure 4a.

Reviewer comment

Figure 2. Put in the figures 7 days, 21 days, 35 days, and 49 days instead of a, b, c, and d for an easier visualization of the figure. Give the conditions of pH and temperature used. In the system S4, how can it be explained that after 35 days KB2 is absent and B1 is present and after 49 days it is just the contrary?

Answer

Figure 2 and its caption has been corrected. In the S4 system, the KB4 strain was not observed at day 21 due to the antagonism of the KB2 strain. At the same time, the KB2 strain has no negative effect on the B1(2015b) strain that persisted in the culture. The KB2 strain is characterized by the fact that it does not survive in a system with a well-developed B1(2015b) culture. Hence its presence was no longer observed on the 35th day. This allowed the population of the KB4 strain to recover, which was observed again on day 49 (on days 21 and 35, its amount was below detection). On the other hand, the B1(2015b) strain on day 49 was below the detection limit. Hence the strain is not shown in Figure 2d. We cannot explain this phenomenon, especially since it was the only system in which the B1(2015b) strain was not observed on day 49. Perhaps the metabolites from the decomposition of the KB2 strain caused the inhibition of the growth of the B1(2015b) strain. However, further research is needed to explain this phenomenon. This problem was discussed in more detail in the manuscript.

Reviewer comment

Line 150. Where is the information that these are mesophilic strains?

Answer

The strains we study are mesophilic, confirmed by the literature on them. Appropriate citations have been incorporated into the manuscript.

Reviewer comment

Lines 175-183. This paragraph should be positioned after Table 1 and before the influence of temperature and pH, since it does not belong to the degradation in the presence of co-pollutants (2.1.3.).

Answer

The indicated fragment concerns the degradation of drugs by a strictly defined and selected bacterial consortium (S7 from chapter 2.1.1) under optimal conditions. It cannot be replaced by temperature and pH because the determination of these parameters made it possible to optimize the degradation process. At the same time, Graph 4 is in section 2.1.3. because this system controls the degradation of NSAIDs in the presence of co-contaminants. Paragraph 2.1.3 has been supplemented with relevant information.

Reviewer comment

Figure 4. Where do data in figure 4 come from? Which system? Conditions of pH and temperature? I think that data in figure 4a are given as percentages, and not as mg/l. Taking this into account, data in this figure after 24 days do not match those in Table 1 after 40 days.  In none of the systems studied in Table 1 the percentage of naproxen degraded reached 100%, and for dichlofenac, the maximum degradation reached after 40 days was 40.76%, but in figure 4a it reached 80% in 24 days.

Answer

Figure 4a was indeed in percentage, but we changed it to include concentration units. The graph shows the degradation of drugs by the S7 system selected for further research. The degradation occurred under optimal conditions, determined during the studies on the influence of pH and temperature. These results do not coincide with the degradation by the s7 system from Table 1 because the degradation tests during selecting the best system were not carried out in optimal conditions. The relevant information has been entered into the manuscript.

Reviewer comment

Figure 5. According to Materials and methods section, the initial amounts of drugs used were: naproxen and dichlofenac 1 mg/l, ibuprofen 5 mg/l and paracetamol 10 mg/l, but at time 0 days these amounts are different in each of the different figures (a-i). Explain that.

Answer

At time 0, the NSAID concentration was measured, which is always subject to error. Hence, the values in the graph do not correspond precisely to the values entered. Error bars have been introduced in the chart, and the values fall within these ranges.

Reviewer comment

I think that is necessary to insert a figure with data in the absence of the co-pollutants.

Answer

Figure 4a shows the degradation of drugs without co-contaminants.

Reviewer comment

Line 205. …except for paracetamol.

Answer

Corrected as required.

Reviewer comment

Line 224. …naproxen´s and dichlofenac´s decomposition…

Answer

Corrected as required.

Reviewer comment

Lines 230-231….”the decomposition of paracetamol and ibuprofen were inhibited only in the presence of copper”. That is not true for ibuprofen, which is inhibited also by CR, Cu, phenol, benzoate and ethanol.

Answer

That's right. Thank you for your attention. This sentence has been corrected by removing the word "only".

Reviewer comment

Line 242. 2.2. Degradation studies in a SBR bioreactor system

Answer

Corrected as required.

Reviewer comment

Figure 6. NSAIDs degradation by bacterial consortia in SBR bioreactor (a) and values of…..throughout the process (b).

Answer

Corrected as required.

Reviewer comment

Materials an Methods should be written in order, beginning with the degradation experiments and giving details about in which samples are applied the different techniques.

Answer

Corrected as required.

Reviewer comment

Line 3.3.1. …and stenotrophomonas maltophilia KB2?

Answer

Corrected as required.

Reviewer comment

In which experiments are the fluorescent bacteria used?

Answer

Labelled bacteria were used in all experiments to track species composition.

Reviewer comment

Only when the conditions have been selected?

Answer

Labelled bacteria were always used because we were interested in their survival in mixed cultures. Without marking, it would not be possible to determine the qualitative composition of the tested systems, as in Fig. 4b.

Reviewer comment

Have you a parallel culture with and withot fluorescence?

Answer

No systems without tagged strains were used.

Reviewer comment

4-chlorophenol, DMSO and ZnII have not been used as co-pollutent.

Answer

Thank you for this note. Indeed, no results were presented for the compounds designated. The manuscript has been corrected.

Reviewer 3 Report

The current study examined the implications of a specific vaccine in the degradation management of medical waste (paracetamol and NSAIDs). The topic is relevant and interesting, with a good contribution to an improved future management of medical waste. However, a few changes are required in order to improve the present form of the paper. Specific requirements are listed below:

  1. Abbreviations are explained when they first appear in the main text, even if they have been included in the abstract, and contribute to making the text easier to read and the information conveyed more efficiently. Once an abbreviation has been established and explained, it will be used throughout the entire manuscript, with the exception of the abstract, where it must be treated separately (i.e. in the abstract, NSAIDs (L12) and SBR (L17) reactor should be explained. SBR in the main text, etc.). Please review the whole manuscript in terms of abbreviations.

2.       L25-38 and L205-L222 There is too much information included in a long paragraph, which makes the information difficult to read/understand. Please divide the information into smaller and more understandable paragraphs.

  1. It is advisable to briefly present in the introduction section a few details about the consumption data/mechanism of action/structure of NSAIDs. I suggest checking and referring to: https://doi.org/10.37358/RC.17.9.5826  
  1. The aim of the article should be clarified in the last paragraph of the introduction section. The authors have described what they have done in the study, but they have not presented the precise aim of the study, the novelty/special aspects it brings to the field, or the reason for choosing this topic. What was done in the research is already in the manuscript and is also provided in other sections.
  1. Since the percentage is listed for all values of the 4 molecules (Table 1), it can be moved to the top of the table and not appear at each value.
  2. L223-241 There is no need for blank spaces between paragraphs. Please revise.
  3. Please revise the numbering of the subsections of Chapter 3 because they are confusing and not consecutive.
  4. L410- The authors provided the name, version, country, and city for all software used in the different analyses, but the link and date of access should be included as a bibliographic reference in the references section.
  5. It is better that the section Results and Discussions be dealt with separately in two sections, as in the template provided by the journal, in order to be as clear as possible. The results section will contain only the results of the research, without any bibliographic references, while the discussion section can refer to other studies and should include the limitations of the present study and how they can be addressed by future research directions.

Author Response

Dear Reviewer 3,

Thank you very much for your work and valuable comments. We have corrected the manuscript according to your suggestions. All introduced changes are marked in the file through the change tracking function. Moreover, all changes and explanations are listed below.

We will be very grateful if you accept our corrections.

Sincerely,

Urszula Guzik and other Co-authors

Reviewer comment

Abbreviations are explained when they first appear in the main text, even if they have been included in the abstract, and contribute to making the text easier to read and the information conveyed more efficiently. Once an abbreviation has been established and explained, it will be used throughout the entire manuscript, with the exception of the abstract, where it must be treated separately (i.e. in the abstract, NSAIDs (L12) and SBR (L17) reactor should be explained. SBR in the main text, etc.). Please review the whole manuscript in terms of abbreviations.

Answer

Corrected as required.

Reviewer comment

L25-38 and L205-L222 There is too much information included in a long paragraph, which makes the information difficult to read/understand. Please divide the information into smaller and more understandable paragraphs.

Answer

Corrected as required.

Reviewer comment

It is advisable to briefly present in the introduction section a few details about the consumption data/mechanism of action/structure of NSAIDs. I suggest checking and referring to: https://doi.org/10.37358/RC.17.9.5826 

Answer

Corrected as required. Manuscript has been supplemented about this information.

Reviewer comment

The aim of the article should be clarified in the last paragraph of the introduction section. The authors have described what they have done in the study, but they have not presented the precise aim of the study, the novelty/special aspects it brings to the field, or the reason for choosing this topic. What was done in the research is already in the manuscript and is also provided in other sections.

Answer

According to the reviewer's comment, the purpose of the work has been rearranged.

Reviewer comment

Since the percentage is listed for all values of the 4 molecules (Table 1), it can be moved to the top of the table and not appear at each value.

Answer

Corrected as required.

Reviewer comment

L223-241 There is no need for blank spaces between paragraphs. Please revise.

Answer

The formatting used in the paper was imposed in the journal template. For this reason, we have left a space between paragraphs.

Reviewer comment

Please revise the numbering of the subsections of Chapter 3 because they are confusing and not consecutive.

Answer

Corrected as required.

Reviewer comment

L410- The authors provided the name, version, country, and city for all software used in the different analyses, but the link and date of access should be included as a bibliographic reference in the references section.

Answer

The software used is an application installed on the computer's hard drive. Therefore it is not possible to enter a link and date of access to the software in the bibliography.

Reviewer comment

It is better that the section Results and Discussions be dealt with separately in two sections, as in the template provided by the journal, in order to be as clear as possible. The results section will contain only the results of the research, without any bibliographic references, while the discussion section can refer to other studies and should include the limitations of the present study and how they can be addressed by future research directions.

Answer

The journal allows authors to combine the Results and Discussion sections into one. The authors used this opportunity due to the more accessible discussion of the obtained results based on the literature. This way, referring to the same results twice without losing informativeness was avoided.

Round 2

Reviewer 3 Report

The authors responded to my requests.